# HTL/KAI2 signaling substitutes for light to control plant germination

Jenna E. Hountalas[1]☯, Michael Bunsick[1]☯, Zhenhua Xu[1]☯, Andrea A. Taylor[1], Gianni Pescetto[1], George Ly[1], François-Didier Boyer[2], Christopher S. P. McErlean[3], Shelley Lumba[1,4]*

1 Department of Cell & Systems Biology, University of Toronto, Toronto, Canada, 2 Université Paris-Saclay, CNRS, Institut de Chimie des Substances Naturelles, Gif-sur-Yvette, France, 3 School of Chemistry, The University of Sydney, S'ydney, Australia, 4 Centre for the Analysis of Genome Evolution and Function, University of Toronto, Toronto, Canada

☯ These authors contributed equally to this work.
* shelley.lumba@utoronto.ca

**Data Availability Statement:** The transcriptome data is stored under GEO accession number GSE161704. Python script can be accessed from https://github.com/maplexuci/stringtie_gene_id_replacement.

## Abstract

Plants monitor multiple environmental cues, such as light and temperature, to ensure they germinate at the right time and place. Some specialist plants, like ephemeral fire-following weeds and root parasitic plants, germinate primarily in response to small molecules found in specific environments. Although these species come from distinct clades, they use the same HYPOSENSITIVE TO LIGHT/KARRIKIN INSENSITIVE 2 (HTL/KAI2) signaling pathway, to perceive different small molecules suggesting convergent evolution on this pathway. Here, we show that HTL/KAI2 signaling in *Arabidopsis thaliana* bypasses the light requirement for germination. The HTL/KAI2 downstream component, SUPPRESSOR OF MAX2 1 (SMAX1) accumulates in the dark and is necessary for PHYTOCHROME INTERACTING FACTOR 1/PHYTOCHROME INTERACTING FACTOR 3-LIKE 5 (PIF1/PIL5) to regulate hormone response pathways conducive to germination. The interaction of HTL/KAI2 and light signaling may help to explain how specialist plants like ephemeral and parasitic weeds evolved their germination behaviour in response to specific environments.

## Author summary

Specialist plants have convergently evolved to use the same family of receptors to perceive different small molecules in lieu of necessary germination cues, like light. Although light is necessary for the germination of many generalist plants such as the model organism *Arabidopsis thaliana*, we show that activating the conserved receptor in *Arabidopsis* promotes germination in the absence of light. We show that the downstream component of the *AtKAI2* signaling pathway is stable in the dark which explains its ability to negatively regulate germination. Additionally, we show that activating the receptor in the dark can modulate the gene expression profile by regulating key components of the light signaling pathway. Our study highlights how specialist plants may recycle components of conserved signaling pathways to thrive in new environments.

**Funding:** This work was supported by the Social Sciences and Humanities Research Council (SSHRC) New Frontiers in Research Fund Exploration program (2018-00118) and the Natural Sciences and Engineering Research Council of Canada (NSERC) in the form of a Discovery Grant (06752), an Accelerator Supplement (507992) and a Research Tools and Instruments grant awarded to S.L. Z.X. received a salary from SSHRC New Frontiers (2018-00118), NSERC (06752) and an NSERC Accelerator Supplement (507992). J.E.H, M.B., Z.X., A.A.T., G.P, and G.L received a salary from NSERC (06752) and an NSERC Accelerator Supplement (507992). The funders did not play a role in the study design, data collection and analysis, decision to publish, or preparation of the manuscript.

**Competing interests:** The authors have declared that no competing interests exist.

## Introduction

Germination strategies are key to determining the environment that a plant will experience, thus, control of this developmental process is highly adaptive [1]. Generally, many plants monitor multiple environmental cues such as temperature changes, moisture content, and light quality to determine when to germinate. By contrast, a single cue dominates germination in some specialist species. For example, seeds of some pyroendemic species remain dormant and only germinate when exposed to a collection of structurally related small molecules called karrikins (KARs), which are released from charred plant matter after a fire [2,3]. Parasitic plants of the *Striga* genus are another example. These species only germinate when they detect the phytohormone strigolactone (SL) released by a nearby host plant [4,5]. Intriguingly, these pyroendemic and parasitic plants perceive these germination cues through a conserved receptor class designated, *HYPOSENSITIVE TO LIGHT/KARRIKIN INSENSITIVE 2* (*HTL/KAI2*) [6–8]. However, since these species are not amenable to genetic analysis, it is difficult to dissect how activation of the HTL/KAI2 receptor elicits a germination response.

Fortunately, HTL/KAI2 signaling also occurs in model experimental plants like *Arabidopsis thaliana* (*Arabidopsis*). The HTL/KAI2 receptor in *Arabidopsis* (AtKAI2) interacts with an F-box protein, MORE AXILLARY MERISTEMS 2 (MAX2) to degrade two negative regulators: SUPPRESSOR OF MAX2 1 (SMAX1) and SUPPRESSOR OF MAX2 LIKE 2 (SMXL2) [4,5]. Presently, the ligand for AtKAI2 is not identified although this receptor is closely related to other α/β hydrolase receptors that bind the hormone strigolactone (SL) [7–9]. Chemically, canonical SL structures are tricyclic lactones (ABC rings) connected to a butenolide ring (D-ring) via an enol-ether bridge. The stereochemistry around the D-ring is important as only SLs with (+)-(2'R) stereochemistry are bioactive [10]. Surprisingly, unlike SL receptors, AtKAI2 is activated by SLs with (-)-(2'S) stereochemistry [10]. However, because 2'S isomers do not occur in nature, a compound, dubbed KL, for KAI2-ligand, has been hypothesized [9]. More perplexing, AtKAI2 is activated by KARs although *Arabidopsis* is not a fire-follower. Although loss-of-function mutations in *AtKAI2* result in increased primary dormancy in some *Arabidopsis* accessions, *AtKAI2* at first glance, appears to play only a minor role in germination of after-ripened seed [11,12].

The first loss-of-function mutant isolated in *AtKAI2* was termed hyposensitive to light (*htl*), indicating the *KAI2/HTL* pathway is a positive regulator of light responses [13]. While germination of many plant species is not affected by light, germination of photoblastic seed is affected by light [14, see 15 for review]. In the case of *Arabidopsis*, which physiologically establishes seed dormancy, light positively promotes germination, an example of positive photoblasticism [see 15,16 for review]. There are, however, negative photoblastic plant species such as onion and lily for which light inhibits germination [14, see 15 for review]. Although the molecular mechanisms underlying this distinction in seeds are unclear, light could be alleviating or promoting the repression of the final set of genes involved in germination in positive and negative photoblastic species, respectively [17–19]. Intriguingly, KARs can substitute for light in germinating the positive photoblastic lettuce species, *Lactuca sativa* cv. Grand Rapids [20–22]. KAR appears to act through a KAI2 receptor in lettuce (LsKAI2b) based on heterologous expression of *LsKAI2b* in *Arabidopsis* [23]. The sufficiency of the KAI2 pathway in lettuce strongly suggests that the KAI2 pathway can replace the light requirement for germination in some positive photoblastic species. To elucidate the mechanisms by which the KAI2 pathway substitutes for light, we investigated this ability in a model positive photoblastic plant, *Arabidopsis thaliana*.

For *Arabidopsis* seed germination, the phytochrome photoreceptors play the biggest role with PHYTOCHROME B (PHYB) promoting germination of imbibed seed in red light

(660 nM) and as seeds hydrate PHYTOCHROME A (PHYA) contributes to a positive far-red (700 nM) light signal [24,25]. Light-activated PHYB and PHYA, in turn, inactivate a key basic helix-loop-helix (bHLH) transcriptional regulator PHYTOCHROME-INTERACTING FACTOR 1 (PIF1; also known as PHYTOCHROME INTERACTING FACTOR 3-LIKE 5 (PIL5) and BHLH105) [26–28]. A reduction in PIF1 action promotes germination through the accumulation of positive regulators of light responses [29].

In the seed, PIF1 modulates other functions including the levels of two key hormones, gibberellins (GA), which promote germination and abscisic acid (ABA), which inhibits it [26,27,30]. PIF1 achieves this by transcriptionally activating of *SOMNUS* (*SOM*), which encodes a CCCH type zinc finger protein [31]. Active SOM downregulates GA biosynthetic genes, *GIBBERELLIN3-OXIDASE 1* (*GA3ox1*) and *GIBBERELLIN3-OXIDASE 2* (*GA3ox2*) and upregulates the GA turnover gene, *GIBBERELLIN2-OXIDASE 2 (GA2ox2)* [31–33]. Concurrently, SOM inhibits expression of a gene encoding an enzyme for ABA breakdown, *CYTOCHROME707-A2* (*CYP707A2*) and upregulates *ABA DEFICIENT 1* (*ABA1*), *NINE-CIS-EPOXYCAROTENOID DIOXYGENASE6* (*NCED6*) and *NINE-CIS-EPOXYCAROTENOID DIOXYGENASE 9* (*NCED9*), which are genes involved in ABA synthesis [31–33]. PIF1 also induces the expression of several genes involved in cell wall loosening, allowing the embryonic root (radicle) to push through a weakened seed coat (testa) [29]. The overall importance of PIF1 in germination is reflected in the ability of *pif1* loss-of-function mutant seed to germinate in the dark in contrast to wild-type seed [26].

The *HTL/KAI2* signaling pathway adds complexity to the interactions among light, GA and ABA signaling in germination. Double mutants between *pif1* and either *Atkai2* or *max2* show intermediate germination phenotypes, suggesting KAR signaling acts independently of PIF1 in *Arabidopsis* [34,35]. Transcriptome studies, however, reveal that many KAR-induced genes are also induced by red light [27]. Moreover, *Atkai2* and *max2* seed exhibit a reduced seed germination under red light while *smax1* mutants show enhanced germination [36–38]. Furthermore, *Atkai2* mutant seed show altered expression of PIF1-regulated genes such as *GA3ox1*, *GA2ox2*, *ABA1*, and *NCED9* suggesting KAR signaling is linked to the PHYB-PIF1 signaling module [35]. Related to this, SMAX1 binds the promoter of the *GA3ox2* repressing its transcription and the SMAX1 protein interacts with GA dependent repressors, RGA-LIKE 1 (RGL1) and RGA-LIKE 1 (RGL3) [38,39]. By contrast, in the light, loss of SMAX1 and SMXL2 function allows *Arabidopsis* to germinate without GA action suggesting the AtKAI2/HTL pathway is independent of GA signaling under these conditions [40]. These conflicting observations most likely reflect the complex roles of light on many aspects of plant growth and development.

The observation that the HTL/KAI2 signaling pathway appears to have a role in light signaling suggested to us that designing experiments under conditions that limit light exposure may have utility in understanding the role of this signaling pathway in germination. Here, we find that activating KAI2/HTL signaling in the dark affects numerous PIF1-regulated genes, including genes involved in ABA and GA synthesis as well as genes that modify cell wall composition. Moreover, like *pif1* mutant seed, we find loss of SMAX1 and SMXL2 function also allow seeds to germinate in the dark. Genetic analysis suggests SMAX1 and SMXL2 act through PIF1 to inhibit germination and this relationship posits how KAR and SL could replace light as a germination cue in certain plant species. This model has implications on how specialist plants like some ephemeral weeds and parasitic plants may have evolved their germination behaviours in response to specific small molecule germination cues.

## Results

### SMAX1 and SMXL2 inhibits *Arabidopsis* germination in the dark

The conditions used to germinate *Arabidopsis* vary widely in the literature. Typically, after-ripened seeds are exposed to various sterilizing agents and then plated onto rich nutrient agar plates with or without a sugar source. Moreover, many dark-based experiments involve pre-exposing seeds to a complicated light regime involving red and far-red light exposure before germination to ensure the ratio of active ($P_{fr}$) to inactive phytochrome ($P_r$) is experimentally correct. To analyze the germination of various genotypes and perform large-scale mutant screens under light-limited conditions, we simplified dark germination conditions of treated seeds in the following way.

First, we removed the confounding effects of seed age on levels of dormancy by only using seeds that had ripened for at least six months; a timeframe based on observations that at least three months of seed ripening is required to maximally decay PHYB to its $P_r$ form [41]. We tested this by giving seeds a five-minute pulse of far-red light and found similar germination responses to our dark conditions (**S1 Fig**). Second, seeds were plated on water and agar media devoid of nutrients and a carbon source, which removes the promotive effects of nitrogen and carbon on germination. Finally, we were concerned that seed sterilization may influence seed germinability in the dark and tested the effects of no sterilization, ethanol sterilization, and bleach sterilization on the germination of seeds in the dark (**S2A Fig**). We found unsterilized or ethanol-sterilized seeds of wild-type (Col-0) reproducibly did not germinate under our dark conditions (**S2A Fig**). Bleaching seed, however, markedly increased the germination of these genotypes in the dark, which most likely reflects a weakening of the seed coat due to the bleach (**S2A Fig**). Because experiments would now require unsterilized seeds, we added the fungicide benomyl to the media and found this has no obvious effect on germination or seedling viability (**S2B Fig**). Based on these results, our dark germination experiments of plating six-month-old dry unsterilized seeds directly onto water agar plates followed by immediate placement in the dark is a useful and simple method of testing dark germination response in *Arabidopsis*.

To further characterize wild-type seed germination under light-limiting conditions, we exposed imbibed seeds to a low light fluence ($\sim$0.3 μmol m$^{-2}$ s$^{-1}$) for increasing time periods to determine a threshold of light duration required for germination. Wild-type seeds showed a slow, steady rise in germination as low-light exposure progressed. In contrast, *htl-3* (*kai2*) seed germinated poorly over the same timeframe, suggesting that the AtKAI2 pathway positively contributed to germination under limiting light conditions, which is consistent with the original identification of KAI2 as 'hyposensitive to light' at the level of inhibition of hypocotyl growth (**S2C Fig**) [13]. Two additional genetic experiments further supported this premise. First, two well-characterized *htl-3* lines overexpressing *AtKAI2* (*AtKAI2OX-A*, *AtKAI2OX-B*) germinated in the absence of light at effective KAR$_2$ concentrations (EC$_{50}$) in the low nanomolar range (**Fig 1A** and **1B**) [8]. Second, genetic inactivation of the downstream negative regulator of AtKAI2 signaling, *SUPPRESSOR OF MAX2 1* (*SMAX1*) resulted in partial germination in the dark (**Fig 1C**). When a loss-of-function mutation in *SUPPRESSOR OF MAX2 LIKE 2* (*SMXL2*), a partially redundant ortholog of *SMAX1*, was introduced into this line (*smax1-2; smxl2-1*), almost all seeds germinated constitutively in the dark (**Fig 1C**). Consistent with our genetic results, the expression of genes known to be induced by *AtKAI2* activation (*DLK2*, *KUF1*, *BBX20*) increased in dark-germinated *AtKAI2OX-A* seeds treated with KAR$_2$ (**Fig 1D**) [42,43]. As predicted, these genes were constitutively expressed in *smax1-2* single and *smax1-2; smxl2-1* double mutants (**Fig 1D**). Our results indicate that SMAX1 and SMXL2 contribute to the repression of germination of *Arabidopsis* seeds in the dark.

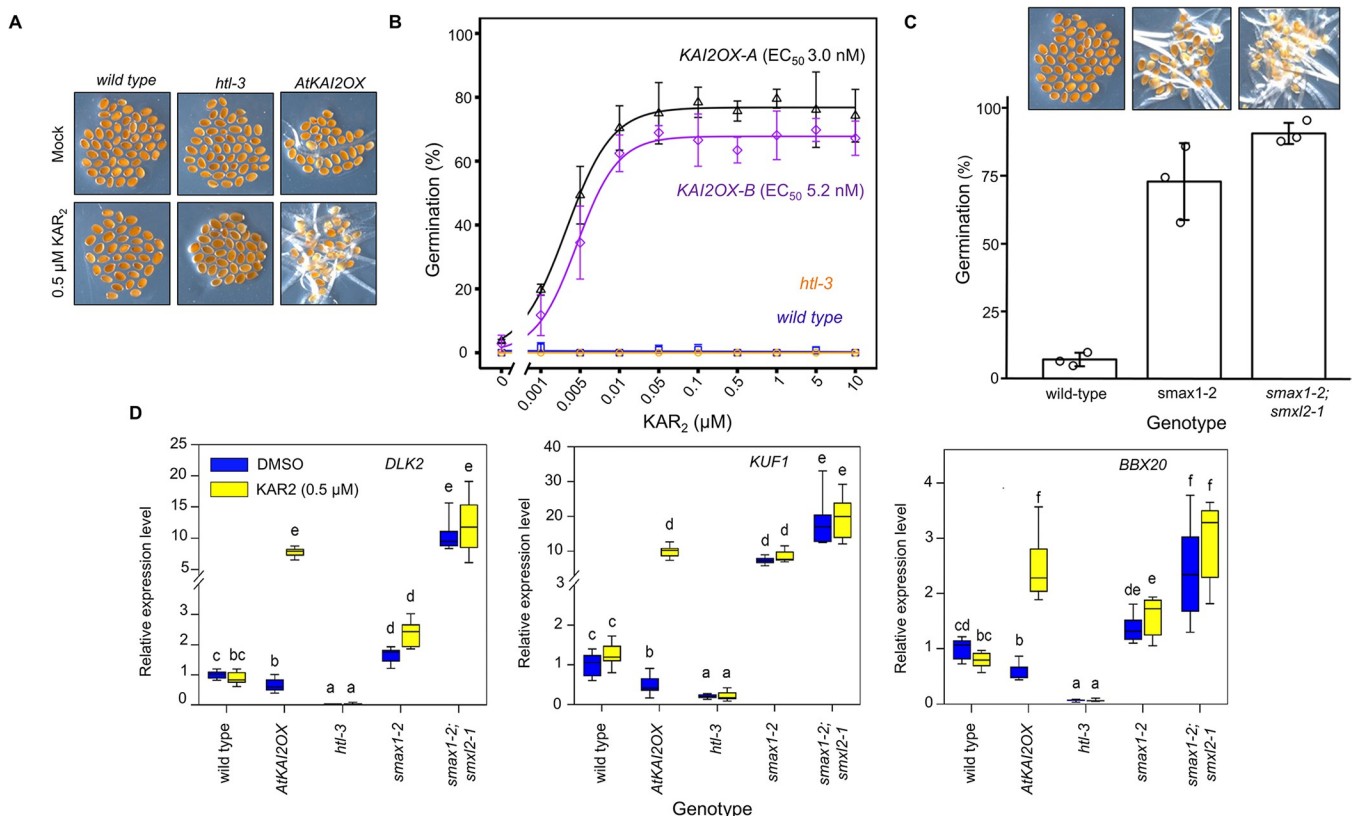

**Fig 1. Activation of AtKAI2 signaling promotes *Arabidopsis* germination in the dark. (A)** Representative images of *Arabidopsis* genotypes germinating on minimal mock media (DMSO) or media containing 0.5 μM KAR$_2$ under continuous dark conditions. **(B)** Effective concentrations (EC$_{50}$) of KAR$_2$ required to germinate 50% of two *AtKAI2* overexpression lines *(AtKAI2OX-A, AtKAI2OX-B)* [8], wild-type Col-0, and *htl-3* seed. Each point represents the mean germination percentage of three biological replicates. Bar = SD. **(C)** Dark germination of Col-0, *smax1-2*, and *smax1-2; smxl2*-1 double homozygous mutant. Each open circle represents the germination percentage of a biological replicate. Bar = SD **(D)** Expression (RT–qPCR analysis) of KAR-induced genes (*DLK2*, *KUF1*, and *BBX20*) [42,43] in dark-germinated *AtKAI2OX-A* seeds on 0.5 μM KAR$_2$. *AtACTIN8* gene was used as an internal control. Lower case letters indicated a significant difference compared to untreated (DMSO) control seeds ($P < 0.05$, ANOVA with post-hoc one-sided Fisher's least significant difference). *P* values, sample sizes and plot elements are provided in S1 Table.

## Light reverses SMAX1 accumulation in the dark

In *Arabidopsis*, SMAX1 and SMXL2 are degraded by the F-box protein, MAX2, when AtKAI2 is activated by either the (-)-(2'S) isomer of *rac*-GR24 or by KARs [44–46]. Evidence for this model, however, often requires special experimental conditions because of difficulties in detecting SMAX1 protein in *Arabidopsis* [44]. Our result that SMAX1 is a negative regulator and represses germination in the dark led us to hypothesize that the SMAX1 protein in *Arabidopsis* should accumulate in the dark. To test this, we fused *eYFP* (enhanced yellow fluorescent protein) to *SMAX1* and placed it under the control of the ubiquitously expressed *35S* promoter (*35S::eYFP-SMAX1*). We generated stable transgenic lines in a *smax1-2* loss-of-function mutant background and found that the *eYFP-SMAX1* transgene complemented *smax1-2* phenotypes, such as hypocotyl length and germination in the presence of the GA biosynthesis inhibitor, paclobutrazol (PAC) (**S3 Fig**).

In contrast to seeds germinated in the light, dark-grown seedlings showed a strong nuclear-localized eYFP signal throughout the root (**Fig 2A**). Importantly, the dark-dependent SMAX1 accumulation disappeared within six hours of light exposure (**Fig 2A**). When we performed the reciprocal experiment, where light-grown seedlings were moved to the dark, we observed

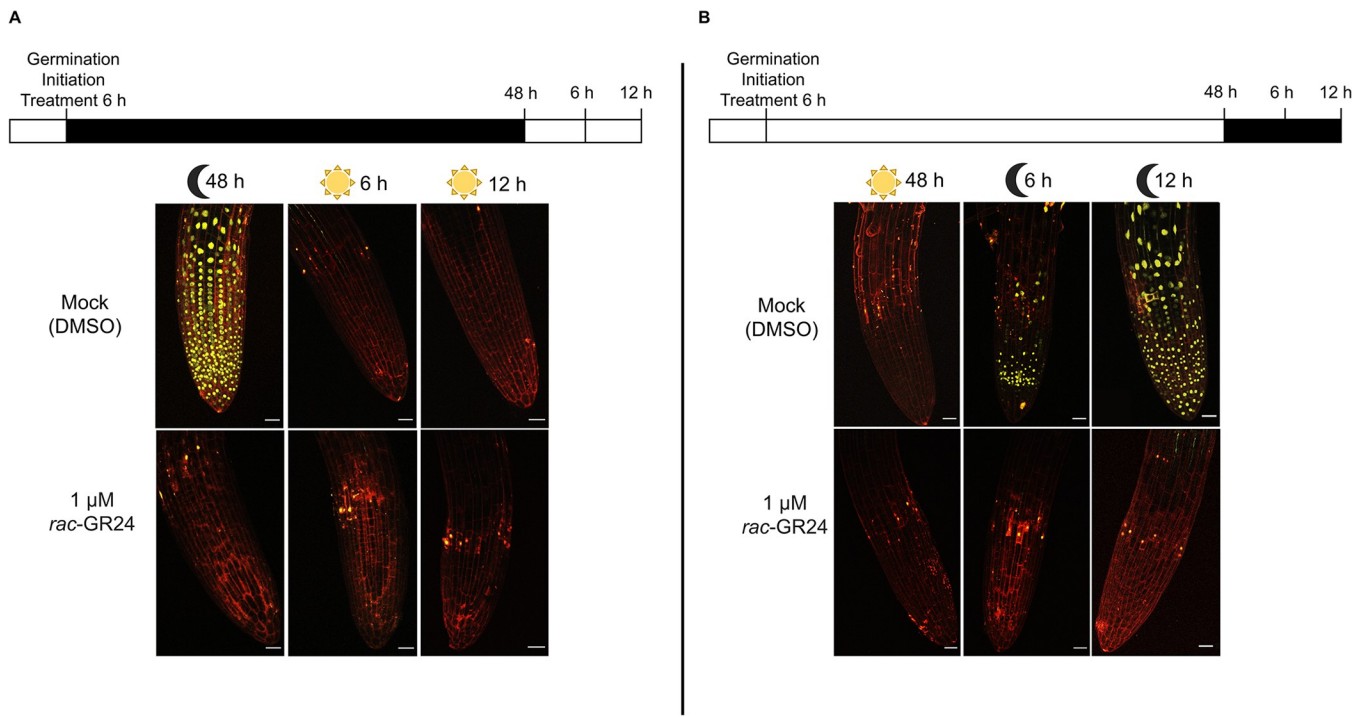

**Fig 2. SMAX1 accumulates in the dark. (A)** Representative z-stack projection images of an overexpression *eYFP-SMAX1* line (*smax1-2; 35S::eYFP-SMAX1*). Seeds were exposed to light for six hours to initiate germination and then transferred to total darkness for 48 hours. After 48 hours in the dark, seedlings were exposed to light for 6 or 12 hours. The red outline is propidium iodide staining of the cell walls. The duration of light and dark treatments is drawn at the top of the figure. **(B)** Representative z-stack projection images of an overexpression *eYFP-SMAX1* line (*smax1-2; 35S::eYFP-SMAX1)* after 48 hours of continuous light germination with or without 1μM *rac*-GR24. Seeds were exposed to light for 48 hours. After 48 hours in the light, seedlings were placed in the dark for 6 or 12 hours. The red outline is propidium iodide staining of the cell walls. The duration of light and dark treatments is drawn at the top of the figure. Scale bar = 25μm.

nuclear-localized eYFP signal within six hours (**Fig 2B**). This signal accumulated to higher levels after twelve hours in the dark (**Fig 2B**). To assess if the *eYFP-SMAX1* line was responding correctly, we reperformed our light shift experiments in the presence of *rac*-GR24. Exposure to *rac*-GR24 activates *AtKAI2* signaling through the (-)-(2'S) isomer and should result in SMAX1 degradation. Indeed, only weak YFP signal was detected in *rac*-GR24 treated roots under all conditions (**Fig 2A** and **2B).** In summary, SMAX1 accumulates in the dark. Genetic analysis indicates that SMAX1 accumulation represses germination in the absence of light stimulus. Upon exposure to light, the SMAX1 signal disappears. The ability of *rac*-GR24 to prevent SMAX1 accumulation in the dark suggested that SMAX1 stability depends on activation of AtKAI2 signaling.

## SMAX1 regulates PIF1 to mediate hormone action

Although SMAX1 and SMXL2 repress seed germination under dark conditions, it is not clear how these proteins regulate this process. Activation of AtKAI2 is thought to primarily target SMAX1 and SMXL2 proteins for degradation [44–46]. Taking advantage of this specificity, we compared global gene expression changes between untreated and KAR$_2$-treated *AtKAI2OX* seeds imbibed in the dark. Despite the absence of light stimulus, KAR$_2$-treatment of *AtKAI2OX* seed upregulated the expression of many genes typically only induced by light (**Fig 3A** and **S2 Table**) [47]. Seedling greening after germination is a highly regulated process that balances the establishment of photosynthesis whilst minimizing photooxidative damage

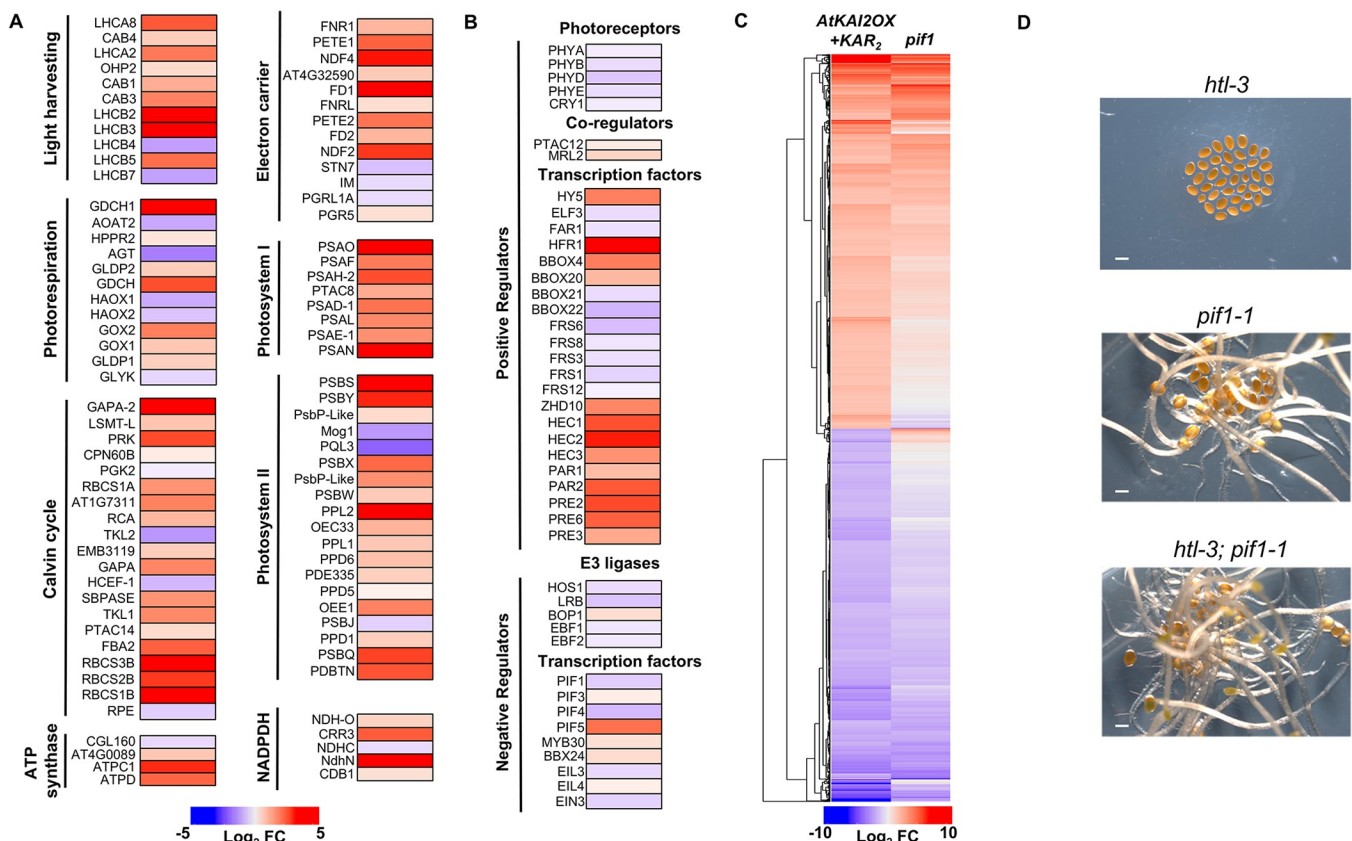

**Fig 3. Activated *AtKAI2* signaling perturbs light signaling. (A)** Heatmap representation of the expression of genes annotated as components of the photosynthetic apparatus in dark germinated *AtKAI2OX* seed treated with KAR₂. All heatmaps depict log₂-transformed fold-change in gene expression between *AtKAI2OX*+KAR₂/*AtKAI2OX* at 24h. See **S2 Table** for gene lists and values. **(B)** Heatmap representation of the expression of genes annotated as regulatory components of phytochrome signaling in dark-germinated *AtKAI2OX* seed treated with KAR₂. **(C)** Heatmap representation of the expression of annotated *PIF1*-regulated genes [29] in dark-germinated *AtKAI2OX* seed with KAR₂ at 24 hours. See **S3 Table** for gene lists and values. **(D)** Representative pictures of *htl-3*, *pif1-1*, and *htl-3; pif1-1* seed placed under dark conditions for five days.

and many of the involved genes are controlled by phytochrome through inhibition of PIF1 [48]. Consistent with this, while cataloguing components involved in phytochrome signaling, we noticed that expression of several basic helix-loop-helix (bHLH) family members including *PACLOBUTRAZOL RESISTANCES* (*PREs*), *PHYTOCHROME RAPIDLY REGULATED* (*PARs*) and *LONG HYPOCOTYL FAR RED1* (*HFR1*) that contribute to seedling greening, were upregulated in KAR₂-treated *AtKAI2OX* seed (**Fig 3B** and **S2 Table**). These bHLHs are thought to fine-tune phytochrome responses by forming feedback loops where the transcription factor, PIF1, promotes *HFR1*, *HECATES* (*HECs*), and *PAR* expression [49–51]. In turn, the HFR1, HECs, and PAR proteins sequester PIF1 to prevent transcription of downstream targets [49–51]. Notably, 94 of the 166 known direct targets of PIF1 were activated two-fold by KAR₂ (**Fig 3C** and **S3 Table**). [29]. This large overlap of gene expression together with the shared dark germination phenotype of *pif1-1* and *smax1-2* mutants, suggest that a genetic relationship exists between these pathways. Indeed, epistatic analysis between *htl-3* and *pif1-1*, which germinates poorly and constitutively under light-limiting conditions, respectively, places *PIF1* at or downstream of *AtKAI2* in a genetically dependent pathway (**Fig 3D**).

Since many of the genes downstream of *PIF1* are annotated as responsive to hormones, PIF1 appears to contribute to establishing the correct combination of hormonal responses

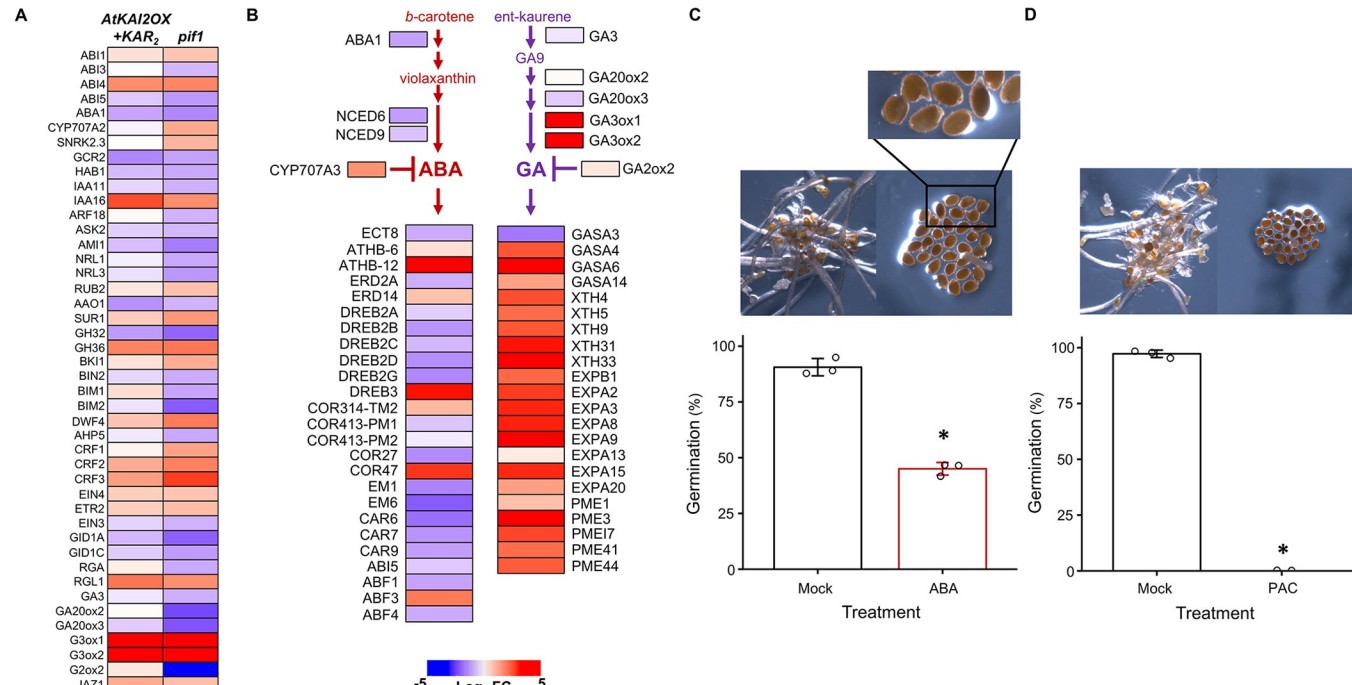

**Fig 4. Activating *AtKAI2* signaling perturbs hormone signaling.** (A) Heatmap representation of the expression of genes annotated as PIF1-regulated hormonal response genes [29] in dark-germinated *AtKAI2OX* seed with KAR$_2$ relative to mock treated (*AtKAI2OX*+KAR$_2$/*AtKAI2OX*). All heatmaps depict log$_2$-transformed fold-change at 24h. See **S4 Table** for gene lists and values. (B) Heatmap representing select ABA-and GA-regulated genes in dark germinated *AtKAI2OX* seed with KAR$_2$. Dark red pathways are ABA-regulated genes. Purple pathways are GA-regulated genes. Heat map values are the ratio of *AtKAI2OX* +*KAR$_2$*/*AtKAI2OX* gene expression after 24 hrs. See **S2 Table** for gene lists and values. (C) Germination of *smax1-2; smxl2-1* seed in the dark on 3 μM ABA. Each open circle represents the germination percentage of a biological replicate. Bar = SD. (D) Germination of *smax1-2; smxl2-1* seed in the dark on 20 μM PAC. Each open circle represents the germination percentage of a biological replicate. Bar = SD. Asterisks indicated a significant difference compared to untreated (mock) *smax1-2; smxl2-1* seeds (*P* < 0.05, one-way ANOVA with post-hoc Tukey honest significant difference test). *P* values, sample sizes and bar plot elements are provided in S1 Table.

required for germination [29]. We observed that the expression of genes involved in hormone response affected by the loss of PIF1 were also regulated by KAR$_2$ activation of AtKAI2 in dark germinated *AtKAI2OX* seed (**Fig 4A** and **S4 Table**). The ratio of the hormones, ABA to GA is crucial for tuning the seed for germination [52]. For example, PIF1 partially represses germination by increasing ABA and decreasing GA levels through SOM [31]. SOM increases ABA levels by activating ABA anabolic genes, *ABA1*, *NCED6*, and *NCED9* as well as repressing the ABA catabolic gene, *CYP707A2* [31]. SOM also decreases the expression of GA anabolic genes, *GA3ox1* and *GA3ox2* and increases the transcript levels of the GA catabolic gene, *GA2ox2* [31]. We found that KAR$_2$ activation of AtKAI2 in dark-germinated *AtKAI2OX* seed decreased transcript levels of the SOM targets *ABA1*, *NCED6* and *NCED9* (**Fig 4B** and **S2 Table**). AtKAI2 activation also increases the levels of *GA3ox1* and *GA3ox2* transcripts whilst decreasing *GA20ox3* (**Fig 4B**). Unlike *SOM*, however, AtKAI2 activation induced expression of a different *CYP707* homolog (*CYP707A3* versus *CYP707A2*) and decreased expression of *GA2ox2* (**Fig 4B**). Even considering this nuance, the transcript profile of KAR$_2$ activation of AtKAI2, demonstrated an overall similarity to the transcript profile of *pif1* loss-of-function seeds. Consistent with a decrease in ABA and increase in GA levels, the expression of genes annotated as ABA-regulated, decreased upon AtKAI2 activation while transcripts of GA-responsive genes increased (**Fig 4A and 4B**).

To functionally test the contributions of ABA and GA to AtKAI2 signaling, we next manipulated the levels of these hormones in the *smax1-2; smxl2-1* double mutant and monitored

dark germination. Adding ABA to *smax1-2; smxl2-1* seeds partially inhibited germination in the dark (**Fig 4C**) as measured by radicle emergence, but ABA did fully inhibit seedling emergence (**S4 Fig**). These phenotypes are consistent with the previous finding that ABA does not increase the DELLA repressors of GA signaling to levels needed to inhibit seed coat rupture [53]. The inability of *smax1-2; smxl2-1* seed to overcome the capacity of ABA to stop seedling greening also suggests that ABA signaling is functional after testa rupture. In contrast to increasing ABA, we found that reducing GA levels by exposing *smax1-2; smxl2-1* seed to paclobutrazol (PAC), a GA biosynthesis inhibitor, fully inhibited germination (**Fig 4D**).

## Identification of dark germination mutants

The ability of the *smax1-2* loss-of-function mutant to germinate in the dark encouraged us to further probe this process by performing an unbiased forward genetic screen to identify new mutations in *AtKAI2* signaling. We screened 48,000 EMS mutagenized Col-0 seeds (48 $M_2$ pools) for mutants capable of germinating under our dark conditions and identified 122 potential mutants (**Fig 5** and **S5 Table**). To speed up the mutant classification, we used a chemical genetics approach where various compounds that are known to differentiate different hormone- and light-related mutants were tested against the mutant collection. We exposed mutant seed to ABA and PAC in the light as these two compounds are commonly used to probe the ABA and GA signaling pathways as they pertain to *Arabidopsis* wild type seed germination. We also included a condition of PAC plus *rac*-GR24 to see if any PAC sensitive mutants could be rescued by addition of this artificial SL/KL. This information was then used to classify groups using two-dimensional hierarchical clustering. The response of a reference set of known hormonal or light signaling mutants to our compounds were also queried through this system to further catalog our dark germination lines. This reference set included

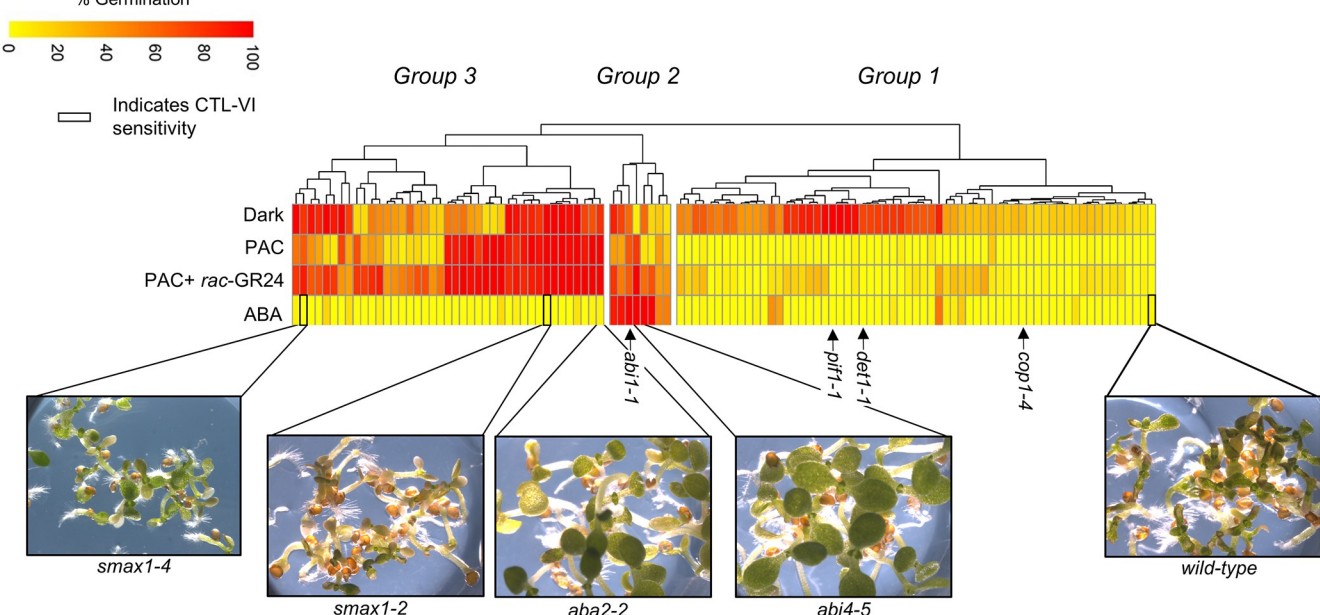

**Fig 5. A genetic screen for dark germination mutants.** Heatmap representation of mutant germination phenotypes on 20 μM PAC, 20 μM PAC, and 1 μM rac-GR24, 2 μM ABA, or in the absence of light. The germination responses were hierarchically clustered using the complete linkage method with Euclidean Distance. The mutants fall into three groups: *det1-1* like (Group 1), *abi1-1* like (Group 2), and *smax1-2* like (Group 3). The photographs show the bleached seedling phenotype of various mutants grown on 2.5 μM CTL-VI. Bolded cells indicate mutants which showed wild-type sensitivity to CTL-VI. *smax1-4* is a novel *SMAX1* allele. See **S5 Table** for putative mutant list and values.

an ABA-auxotroph, *aba deficient2* (*aba2-2*), two ABA-insensitive mutants, *aba insensitive1* (*abi1-1*), *aba insensitive4* (*abi4-5*), three constitutive light signaling mutants, *phytochrome interacting factor1* (*pif1-1*), *de-etiolated1* (*det1-1*), *constitutive photomorphogenic1* (*cop1-4*) and the constitutive *AtKAI2* signaling mutant, *smax1-2*.

Using these criteria, we identified three major groupings. Group 1 mutants did not germinate well on either PAC or ABA, indicating they are not perturbed in GA or ABA signaling (**Fig 5**). Within this group, the light signaling mutants *pif1-1* and *det1-1*, which also germinate in the dark, did not germinate well on PAC or ABA suggesting this group may be enriched for mutants defective in light signaling (**Fig 5**). Group 2 mutants showed moderate germination in the dark, high germination on ABA, and varying germination on PAC (**Fig 5**). As expected, this group included *abi1-1* and *abi4-5* and therefore most likely is enriched for ABA signaling mutants (**Fig 5**). Finally, Group 3 mutants, which included *smax1-2* and *aba2-2* seed, generally germinated well in the dark and on PAC in the presence or absence of *rac*-GR24 but did not germinate on ABA (**Figs 5** and **S4**). Mutants perturbed in KL signaling can be identified using a compound called cotylimide-VI (CTL-VI) [54]. Addition of CTL-VI to germinating *Arabidopsis* seed bleaches their cotyledons and *max2* mutants, which are insensitive to SL or KL signaling are insensitive to this bleaching effect [54]. We, therefore, thought mutants constitutive for KL signaling, like *smax1-2*, may show increased sensitivity to CTL-VI application versus ABA auxotrophs with respect to cotyledon bleaching. Indeed, CTL-VI preferentially bleaches *smax1-2* cotyledons compared to *aba2-2* and *abi4-5* cotyledons (**Figs 5** and **S5**). Interestingly, five mutants from Group 3 showed a similar bleaching phenotype on CTL-VI to that of *smax1-2* and subsequent sequencing for the *SMAX1* locus identified a new *smax1* allele (*smax1-4*) carrying a nonsense mutation in codon 118 (CAA→TAA). The *smax1-4* allele is predicted to produce the smallest truncated product among all the known *SMAX1* alleles [37] and therefore could represent a good null mutant. Importantly, our isolation of a novel allele in SMAX1, *smax1-4*, indicated that our screening conditions for constitutive dark germinating mutants can identify constitutive *AtKAI2* signaling mutants and directly connects the dark germinating phenotype to the *AtKAI2* signaling pathway.

## Discussion

### SMAX1 and SMXL2 inhibits *Arabidopsis* germination in the dark

Generally, for plants like *Arabidopsis*, GA signaling is the major germination promotive pathway while the KL pathway appears to only have a minor role. By contrast, fire-following ephemeral weeds and some parasitic plant species have an absolute requirement for HTL/KAI2 for germination and are not responsive to GA [3,7,8,11]. We believe we have reconciled these species differences by showing that *Arabidopsis* germination like parasitic and ephemeral weeds can also become completely dependent on AtKAI2 signaling under appropriate environmental conditions. Consistent with this, *smax1; smxl2* double mutants show efficient germination under dark conditions. Loss of SMAX1 and SMXL2 function by the activation of AtKAI2 produced a similar transcriptional signature as loss of the light-dependent repressor, PIF1. Epistatic analysis suggested that these genes act in a common signaling pathway. On this note, SMAX1 reduces the inhibitory effect of PHYB on PIF4, which indicates that phytochrome requires activated AtKAI2 signaling to repress PIF function in the seedling [55]. Since our genetic epistasis and transcriptome comparisons demonstrated a functional and mechanistic connection between SMAX1 and PIF1, we predict that SMAX1 and SMXL2 are also necessary for PIF1 function in the dark.

The ability of light to inhibit SMAX1 accumulation could explain some of the previous experimental difficulties involving SMAX1 cell biology. Fluorescence is not detected in

transgenic *Arabidopsis* containing SMAX1-GFP fusions even when the F-box MAX2, which is thought to control SMAX1 degradation, is genetically removed [44]. By contrast, under our dark growth conditions, we detected accumulation of eYFP-SMAX1 *in planta* with the signal diminishing over a few hours in the presence of *rac*-GR24 or light. The ability of light to inhibit SMAX1 accumulation is similar to a previous study that did not detect SMAX1-GFP in light-grown seedlings unless the SMAX1 degron motif was deleted [44]. However, our results differed from SMAX1 accumulation studies in the hypocotyl, which conclude that light encourages SMAX1 accumulation [55]. Perhaps these results reflect differences in the light regime as well as the tissue and developmental stage used in the studies. In the hypocotyl study, older seedlings were exposed to a short-day light exposure of 8 hours light and 16 hours dark, in contrast to our experiments which were conducted on germinating seeds under constant light or dark. The difference in light exposures could affect the dynamics of SMAX1 accumulation. At different developmental stages, SMAX1 stability could differ in the dark compared to light. Unfortunately, the absence of a KL ligand candidate limits speculation on how light conditions may affect this ligand composition, concentration or sensitivity. However, light and the addition of KARs or *rac*-GR24 stimulate *AtKAI2* transcription, which is consistent with our findings that SMAX1 is degraded in the light during early stages of germination [12,40]. Whatever the case, special consideration regarding the development stage of the plant and light regime should be given when studying SMAX1 stability.

Low nanomolar $KAR_2$ concentrations were sufficient to stimulate high germination rates of *AtKAI2OX* seed in the dark, which was unexpected given that $KAR_2$ addition to *AtKAI2OX* seed in the light only partially germinates seeds depleted of GA [40]. There are numerous reports of GA and SL interactions at the level of synthesis; for example, GA addition negatively regulates the levels of SL in rice [56]. As light is known to increase GA synthesis this would suggest SL levels are lower in the light perhaps sensitizing the *AtKAI2* signaling pathway to exogenous KL [30]. Of course, because KL is not identified, we are not sure this compound is regulated in a similar manner to SL. More directly, SMAX1 interacts with DELLA domain proteins and this interaction appears to enhance SMAX1 transcription and inhibit expression of a gene involved in GA biosynthesis [38]. These relationships support that DELLA and SMAX1 positively enhance one another. Whatever the case, our genetic studies suggest that the SMAX1-DELLA module act together on *Arabidopsis* germination under low light conditions [38].

An alternative explanation for light-dependent SL sensitivities could revolve around the role of PIF proteins [57,58]. Although this interaction appears antagonistic in the hypocotyl, the PIF1-DELLA module appears to be positive in the seed. PIF1 activation reduces GA levels, which in principle would encourage DELLA protein stabilization (**Fig 6**). In this context, PIF1 function also requires SMAX1 and SMXL2, thus inactivation of these proteins through SL signaling increases GA resulting in DELLA protein degradation (**Fig 6**). Accordingly, seed germination in the dark will be PAC dependent. In the light, all PIF proteins are inactivated by phytochromes, consequently DELLA repressors are inactive (**Fig 6**). This light-dependent loss of DELLA function removes the requirement of GA for germination thereby making seed germination insensitive to PAC.

Like all working models there are complexities. For example, differences in light versus dark germination in response to karrikins may also reflect metabolic differences under these environmental conditions. Possibly, KARs are converted to more potent ligands in the dark or perhaps are degraded more effectively in the light. It is important to note that neither the addition of $KAR_2$ nor the overexpression of AtKAI2 was solely sufficient to trigger germination under our dark conditions, which suggests that the expression of *AtKAI2* and production of KL are tightly controlled. Whatever the case, our results demonstrated a major role for

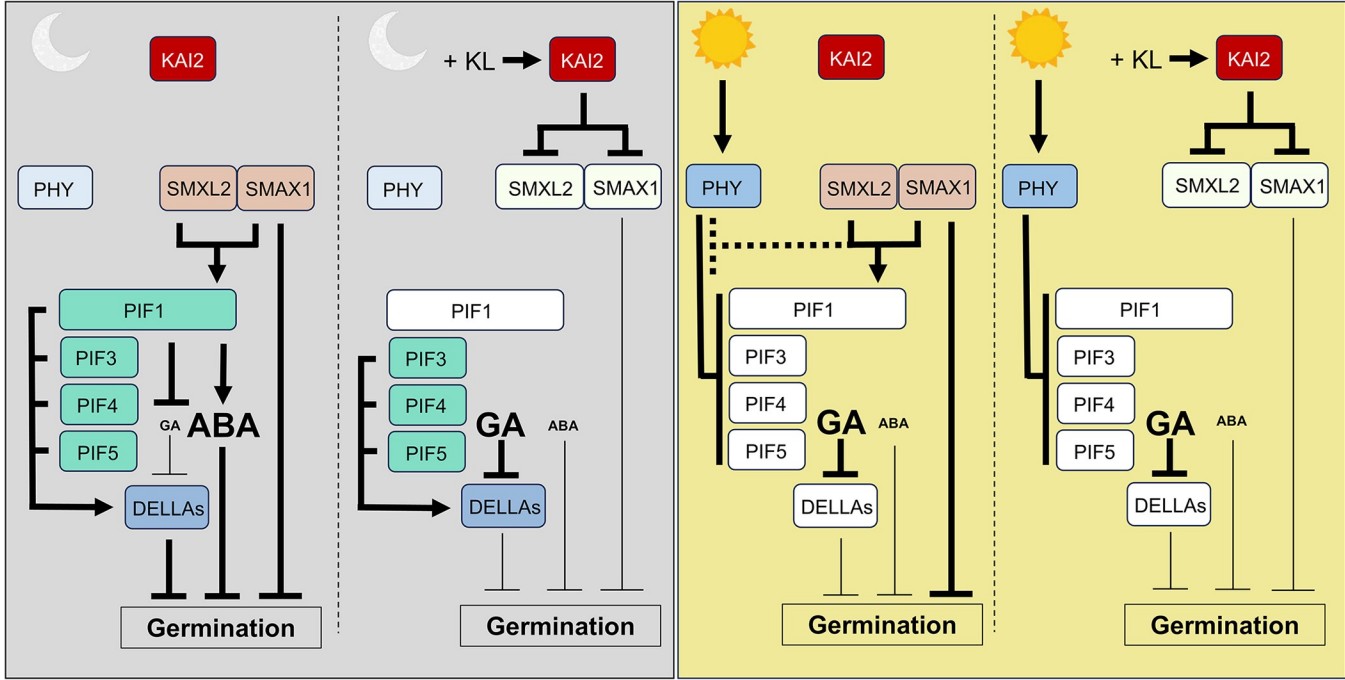

**Fig 6. A working model on the contributions of SMAX1 and SMXL2 to germination under dark versus light conditions.** Far left panel, in the dark, PIF proteins actively repress light response genes. Under low KL conditions, SMAX1 and SMXL2 activate PIF1 to reduce the GA to ABA ratio discouraging germination. SMAX1 and SMXL2 also could act to directly dampen germination as SMAX1 interacts with the EAR motif repressors, TOPLESS1 (TPL1) and TOPLESS RELATED1, 2, 4 (TPR1, 2, 4) [59–62]. Center left panel, increasing KL inhibits SMAX1 and SMXL2 function to both alleviate the repressive effects of these proteins and increase the GA/ABA ratio. Although this allows germination in the dark, inhibition of GA synthesis by addition of the GA synthesis inhibitor, PAC, will decrease the germination response due to the repressive effects of DELLA proteins. Right central panel, in the light, phytochromes inactivate PIF proteins resulting in DELLA protein inactivation. Although seeds no longer require GA to germinate, SMAX1 and SMXL2 proteins still maintain their function to attract TPL and TPR. Also, SMAX1 is thought to inhibit PHYB inactivation of PIF4 in the light. This may be true of other PIF proteins. These events would suggest a weaker germination response in low KL conditions. Far right panel, increased KL levels inhibit SMAX1 and SMXL2 function and this in combination with PIF inactivation of DELLA repressors will lead to a stronger germination response in the presence of PAC.

SMAX1 and SMXL2 signaling in regulating *Arabidopsis* germination. Moreover, we suggest darkness is an optimal environmental condition to study the roles of AtKAI2 signaling in *Arabidopsis* germination.

## Conservation of the roles of HTL/KAI2 signaling in germination

Our analysis gives some insight into the evolution of germination signaling pathways in ephemeral weeds that are fire followers. Adapting a signaling pathway for a particular process broadly follows scenarios where either the pathway is established for its function by natural selection (historical genesis) or the pathway, which has other functions, is co-opted for new uses (current utility) [63]. Fire-following behaviour is a derived trait, thus our findings in *Arabidopsis* suggest the role of *HTL/KAI2* signaling in these species is the product of historical genesis rather than current utility. Our results suggest in generalist plants like *Arabidopsis*, SMAX1 and SMXL2 inhibit germination in the absence of light like PIF1. Unlike PIF1, however, which is modulated by a ubiquitous environmental cue, light, SMAX1 and SMXL2 levels depend on light and specific small molecules generated by distinct environmental conditions. Because SMAX1 and SMXL2 function is necessary to repress dark germination, any selective pressures affecting their levels will dominate germination behaviours.

These findings also suggest that smoke-derived karrikins substitute for light in germinating the positive photoblastic lettuce cultivar, *L. sativa* cv. Grand Rapids, through activation of the KAI2

pathway leading to inactivation of SMAX1 [20–22]. Finally, our results emphasized that defining signaling pathways by their ligands and receptors can be confounding when these signals converge on conserved downstream effectors which are also modulated by other signaling pathways. We suggest when looking at fundamental processes such as germination, it is more informative to reference pathways by their bottom effectors like SMAX1, rather than by specific receptors [64].

## The genetic space of dark germination

The establishment of simple dark conditions creates an experimentally tractable environment to identify genes involved in the germination process. Furthermore, these conditions can be combined with small molecule compounds that perturb germination to quickly classify mutants into categories for subsequent genetic analysis. Traditionally, mutants are classified through genetic complementation, which is tedious in *Arabidopsis*. Using a chemical biology approach to classify mutant lines is based on the concept that each chemical represents a unique environmental condition [65]. With enough compounds and two-dimensional hierarchical clustering, in principle, different classes of mutations can be grouped without performing complementation testing. By focusing on compounds that perturb germination and early seedling growth, we showed the utility of using only four compounds to classify dark germination mutants into three categories, one of which included a new *SMAX1* allele.

In this study, we limited ourselves to chemical mutagenesis which will mostly enrich for loss-of-function mutations in regulators that impede germination under dark conditions but similar experiments with mis-expression collections would identify positive components [66,67]. In our screen, we identified three classes of mutations. DET1, for example, is known to directly interact with and promote the protein stability of PIF1 and our screen placed loss-of-function mutations in DET1 and PIF1 into one class (Group 1) [68]. Mutations in another key light component, COP1 resulted in low germination rates in the dark but this weak phenotype most likely reflects the leakiness of the *cop1-4* allele we used as a reference [69]. Hence, our dark germination screens should enrich for mutants that influence light signaling.

Our screen should also identify new mutations in ABA synthesis and signaling. This may be useful as the identification of mutants in the ABA pathway action often requires screening in the presence of exogenous hormones. Because our dark screen does not require any chemical additions, new alleles may be more relevant to seed physiology. Finally, we expected to identify new negative regulators in AtKAI2 signaling and to this end, we identified a new *SMAX1* nonsense allele. Presently, all the current *SMAX1* alleles either contain nonsense mutations or T-DNA insertions [37].

Our study presents new evidence that activation of the *AtKAI2* signaling pathway can modulate hormone action to germinate in the absence of light. Loss-of-function of the downstream negative regulator, *SMAX1*, can constitutively germinate in the dark and through cell biology experiments, we show that SMAX1 is stable in the dark. Interestingly, the gene expression profile of activated *AtKAI2* signaling reflects that of a light signaling mutant, *pif1*, and we genetically place *PIF1* at or downstream of the *AtKAI2* signaling pathway. Lastly, we utilize the dark germinating phenotype of constitutive AtKAI2 signaling mutants to screen for mutants in this pathway and display the advantages of screening during light-limiting conditions. Therefore, we present novel insights into the *AtKAI2* signaling pathway with respect to the light signaling pathway.

# Materials and methods

## Plant material and plant growth

*Arabidopsis thaliana* Col-0 was the default wild-type (WT) for this study, except for *abi1-1* (*Ler*) [70]. *htl-3* (*kai2*) [71], *smax1-2* (SALK_128579) [37], *smax1-2/smxl2-1* [37], *pif1-1*

[26], *abi4-5* [72], *cop1-4* [72], *aba2-2* [73], *abi1-1* [70], *det1-1* [74], and *Arabidopsis kai2* mutant overexpressing *Arabidopsis AtKAI2* (*35S::AtKAI2*) or *Striga ShHTL7* (*35S::ShHTL7*) [8] were described previously. *Arabidopsis thaliana* plants were grown in a continuous 24-hour light chamber at 22˚C. Seeds were harvested and stored in constant conditions (24˚C in the dark) for after ripening. All genotypes used in this study are listed below and in S7 Table.

## Plasmid construction and plant transformation

The *35S::eYFP* cassette was amplified from pGWB542 and cloned into pGREENII-0179 vector using Fast Digest *KpnI* and *ApaI* restriction enzymes (ThermoScientific, cat no. FD0524, cat no. FD1414) and primers 35S-YFP_F_KpnI:5' CGG<u>GGTACC</u>TGAGACTTTTCAACAAAGGGT 3' and 35S-YFP_R_ApaI:5' GC<u>GGGCCC</u>CTTGTACAGCTCGTCCATG 3'. *Arabidopsis SMAX1* was amplified from *Col-0* cDNA using primers AtSMAX1_F_ApaI: 5' GC <u>GGGCCC</u>ATGAGAGCTGGTTTAAGTACGA 3' and AtSMAX1_R_HindIII: 5' CGG <u>AAGCTT</u>**TCA**TACTGCCAAAGTAATAGTTGT 3' and cloned into pGREENII-0179-eYFP using Fast Digest *ApaI* and *HindIII* (Thermofisher, cat no. FD1414, cat no. FD0504), resulting in the pGREENII0179-eYFP-AtSMAX1 construct. All primers used in this study are listed in S6 Table. The construct was transformed into the *smax1-2* mutant by *Agrobacterium tumefaciens* GV3101- mediated floral dip transformation [75].

## Double mutant construction

Putative *htl-3; pif1* double mutants were identified from $F_2$ offspring of *htl-3; ShHTL7OX* x *pif1* cross by first screening for the *pif1* dark germination phenotype. Plants were then screened for the *htl-3* round rosette leaves and long hypocotyl phenotype. These lines were also screened on glufosinate ammonium (BASTA) to look for lines that were 100% sensitive on BASTA to ensure that the *ShHTL7OX* transgene was absent. Potential *htl-3; pif1* homozygous lines were confirmed by genotyping of individual plants by PCR for the *htl-3* and *pif1* allele. Primers used for genotyping are listed in S6 Table. See S6 Fig for workflow and PCR gels confirming *htl-3; pif1* double mutants.

## Dark germination assays

Seeds for dark germination assays were left to dry for at least six months in a glass tube and closed cardboard storage box at room temperature (24˚C) in the laboratory before use in germination assays. Approximately 25 to 35 of unsterilized dry seeds per replicate were sprinkled directly onto 10 mm petri plates in the dark containing water agar (0.8%) and the fungicide benomyl (10 μg/ml) (Toronto Research Chemicals, B161380). Seeds were exposed to low light (~0.1 μmol m$^{-2}$ s$^{-1}$) for less than 30 seconds while plating and then plates were wrapped with three layers of aluminum foil. Plates were placed at 4˚C for four days and then incubated at 22˚C for five days in the dark. Germination was quantified using radicle emergence. Seedling emergence was quantified by counting fully elongated hypocotyls.

For all chemical treatments, *rac*-GR24 was used which is a racemic mixture of (+)-GR24 and (-)-GR24. A stock solution of *rac*-GR24, PAC (Cayman Chemical Company, 18864), and ABA (Sigma-Aldrich, A1049-100MG) was diluted in DMSO, 100% ethanol, and 100% methanol, respectively. For dark germination assays on compounds, the fungicide benomyl (10 μg/ml) and the appropriate compound at specific concentrations (3 μM ABA, 20 μM PAC) or the solvent were added to the water agar media (0.8%) at 1/1000 concentration.

## Light germination assays

Germination assays on paclobutrazol, cotylimide, and ABA in the light were performed as previously described [40, 54]. Seeds were surfaced sterilized with 70% (v/v) ethanol and gently mixed for five minutes. After the 70% ethanol was removed, 100% ethanol was added, gently mixed for 30 seconds, and then removed. Once dry, approximately 25 to 35 seeds were sprinkled directly onto 10mm petri plates of ½ MS (Murashige-Skoog) agar (0.8%) containing the appropriate compound at specific concentrations or the solvent at 1/1000 concentration. Plates were placed at 4˚C for four days and then incubated at 24˚C for five days in the light. Germination was quantified using radicle emergence. Seedling emergence was quantified by counting fully elongated hypocotyls.

## Far-red germination assays

Approximately 25 to 35 of unsterilized seeds per technical replicate were sprinkled directly onto 10 mm petri plates in the dark containing water agar (0.8%) and the fungicide benomyl (10 µg/ml) (Toronto Research Chemicals, B161380). Plates were wrapped with three layers of aluminum foil and placed at 4˚C for four days. Plates were treated with far-red light at ~5 µmol m$^{-2}$ s$^{-1}$ for 5 minutes and then wrapped in three layers of aluminum foil and incubated at 22˚C for five days in the dark. Germination was quantified using radicle emergence.

## Transcript profiling and data analysis

Total RNA from three biological replicates of *KAI2OX* seeds that germinate in the dark for 24 hours in the presence of 0 µM or 0.5 µM KAR$_2$ was isolated using the mini hot phenol method [76], following the clean-up using RNase-free DNase I and Monarch RNA Cleanup Kit (New England Biolabs) and was used for single-end 1X75bp RNA-Seq on a NextSeq500 sequencing platform (Illumina). After examining the data quality using FastQC (v0.11.5) [77], the fastq data containing the raw reads for each sample was trimmed and filtered using htStream (v1.3.0) (UC, Davis) [78] to generate clean reads. For each read, trimming and filtering processes include removing the last base, removing the low-quality fragments, removing 'N's and returning the longest fragments, and discarding reads that are less than 25 bp. Clean reads from each sample were subjected to HISAT2 [79] (v2.1.0) for genome mapping using *Arabidopsis thaliana* TAIR10 genome as a reference. Samtools (v1.10) [80] was used to remove the unmapped and multi-loci mapped reads and convert the uniquely mapped reads into a BAM file. Transcript assembles for each sample were performed using StringTie (v2.1.3) [81] against *Arabidopsis thaliana* Araport11 [82] annotation and transcriptome assembles from each sample were merged using the *merge* function in StringTie to generate a combined transcriptome across all sequencing samples to serve as a new reference transcriptome. A customized Python script, *stringtieGeneIdReplace2.py* was written to replace the StringTie default 'MSTRG'-tagged *gene_id* with the TAIR10 gene locus name in order to facilitate the comparisons of genes and transcripts between different samples. A StringTie provided Python script, prepDE.py, was used to extract the read counts matrix from each transcriptome assemble and this counts matrix served as the input data for DESeq2 (v1.18.1) [83], an R [84] package for identifying the differentially expressed genes between various comparisons. The transcriptome data is stored under GEO accession number GSE161704. Python script, *stringtieGeneIdReplace2.py* can be accessed from https://github.com/maplexuci/stringtie_gene_id_replacement.

## Gene expression analysis by quantitative RT-PCR

Seed total RNA from three biological replicates of each treatment was isolated using mini hot phenol method [75], following the clean-up using RNase-free DNase I and Monarch RNA

Cleanup Kit (New England Biolabs). One μg of DNA-free total RNA was reverse transcribed using LunaScript RT SuperMix Kit (New England Biolabs) according to the manual. cDNA product was diluted by 10 times and 1 μL was used in the quantitative RT-PCR reaction. The primers used are listed in S6 Table. Real-time PCR was performed using Luna Universal qPCR Master Mix (New England Biolabs) on a Bio-rad CFX96 Real-time Detection System (Bio-rad). The PCR reaction was performed in triplicate for each biological replication. Expression levels were normalized against the expression of the endogenous control gene, *AtACTIN8*, and were calculated using $2^{-\Delta\Delta Ct}$ method [85]. All the expression data was presented as fold change value and was transformed into log2 value for statistical analysis.

## Confocal microscopy

The floral spray technique was used to construct the *smax1-2; 35S::eYFP-SMAX1* line [75]. T1 transgenic lines were selected for resistance on hygromycin and propagated to the T2 generation. Line 1 in the T2 generation was phenotyped for PAC and hygromycin resistance and used for confocal imaging. For imaging of *smax1-2; 35S::eYFP-SMAX1*, seeds were surfaced sterilized with 70% (v/v) ethanol and gently mixed for five minutes. After the 70% ethanol was removed, 100% ethanol was added, gently mixed for 30 seconds, and then removed. Once dry, seeds were sprinkled directly onto 10mm petri plates of ½ MS (Murashige-Skoog) agar (0.8%) in the dark and then wrapped three times in aluminum foil. Plates were placed at 4°C for four days and then exposed to white light (~40 μmol m$^{-2}$ s$^{-1}$) for six hours. Plates were wrapped in three layers of aluminum foil and placed in a room-temperature dark cabinet for 48 hours depending on the light switch experiment and imaged.

Seedlings with specific light treatments were transferred to glass slides with staining solution containing 10μg/ml propidium iodide (PI) and visualized for nuclear localized *e*YFP fluorescence. The Leica TCS SP5 confocal laser microscope was used, and all images are z-stack projections with 3 μm per step overlayed in ImageJ. All images are imaged with the 40x oil immersion objective lens.

## Genetic screen for new *smax1* alleles

To generate an M$_1$ EMS population, approximately 20,000 seeds were mutagenized in 0.2% for 16 hours and then washed in distilled water 10 times before sowing onto soil. For the primary screen, approximately 48,000 M$_2$ seeds from 48 pools of EMS mutagenized *Col-0* were plated onto water-agar plates. The plates were immediately wrapped in three layers of aluminum foil and stratified for four days at 4°C. After stratification, the plates were incubated in the dark at room temperature for five days. Any dark germinating seedlings were then grown to produce M$_3$ seeds. For the phenotypic screening, M$_3$ seeds were plated onto plates containing minimal ½ MS-agar or ½ MS-agar supplemented with either 20 μM PAC, 20 μM PAC + 1 μM GR24, 2 μM ABA, or 2.5 μM CTL-VI. The plates were stratified for four days at 4°C and then incubated under constant white light (~40 μmol m$^{-2}$ s$^{-1}$) at room temperature for seven days. After seven days, seed germination counted by radicle emergence and ABA resistance counted by cotyledon greening was quantified. A heatmap was generated from this data and hierarchically clustered using the Euclidean distance metric in RStudio.

## Statistical analysis

One-way ANOVA analysis was performed using SPSS software (IBM) and Tukey HSD method was used for multiple comparison. Descriptive statistics was perfomed using an online tool CalculatorSoup (https://www.calculatorsoup.com/calculators/statistics/descriptive statistics.php)

Python libraries scipy, pandas and numpy were used in calculating Pearson's correlation coefficient $R^2$, *P*-value and 99% confidence interval.

## Supporting information

**S1 Fig. Activation of the AtKAI2 signaling pathway can germinate seeds in PhyB-off conditions.** (A) Representative images of seeds treated with or without 0.5 μM $KAR_2$ after five minutes of far-red light exposure and placed in a room-temperature dark cabinet. (B) Germination percentages of seeds after a far-red light regime as shown in (A). Germination was counted as radicle emergence. Bar plot represents three biological replicates. Black circles represent the mean of each biological replicate. Bars represent SD.
(TIF)

**S2 Fig. *AtKAI2* is a positive regulator of light germination.** (A) Germination of *Arabidopsis* seed under different sterilization conditions in the dark. (B) Germination percentages in the dark with or without the fungicide, benomyl. Germination was counted as radicle emergence. Bar plot represents three technical replicates. Closed circles represent the mean of each biological replicate. Bars represent SD. (C) Germination under different lengths of exposure to low light (0.3 μmol $m^{-2}$ $s^{-1}$). Sample sizes and box plot elements are provided in S1 Table.
(TIF)

**S3 Fig. *eYFP-SMAX1* overexpression line functionally complements *smax1-2*.** (A) *35S:: eYFP-SMAX1* functionally complements *smax1-2* on 20 μM Paclobutrazol (PAC). Germination on 20 μM PAC of wild type, *smax1-2* and *smax1-2; 35S::eYFP-SMAX1* transgenic line. Average germination was counted for radicle emergence of three technical replicates. Bar = SD. Asterisks indicate a significant difference to wild type (p<0.05, ANOVA one-way with post-hoc Tukey HSD test). (B) Overexpression of *eYFP-SMAX1* is resistant on 20 μM hygromycin. Germination on hygromycin of wild type, *smax1-2* and *smax1-2; 35S::eYFP-SMAX1* transgenic line. Resistance was counted as unwilted cotyledons of three technical replicates. Bars represent SD. Asterisks indicate a significant difference to wild type (p<0.05, ANOVA one-way with post-hoc Tukey HSD test. (C) Overexpression of *eYFP-SMAX1* functionally complements *smax1-2* hypocotyl length. Hypocotyl length of wild-type, *smax1-2*, and *smax1-2; 35S::eYFP-SMAX1* transgenic line. Seedlings were stratified for four days and then grown under white light for seven days. Hypocotyl length was measured with ImageJ. N = 10–11. Representative seedlings displayed above boxplot in order of x-axis. Bar = 1mm. Asterisks indicate a significant difference to wild type (p<0.05, ANOVA one-way with post-hoc Tukey HSD test). *P* values, sample sizes and box plot elements are provided in S1 Table.
(TIF)

**S4 Fig. Abscisic acid (ABA) and paclobutrazol (PAC) treatment inhibits seedling emergence of *smax1-2; smxl2-1* seed.** Seedling emergence was quantified as fully elongated hypocotyls. Black circles represent the mean of each biological replicate. Bar = SD. Asterisks indicate a significant difference compared to untreated (mock) *smax1-2; smxl2-1* seeds ($P < 0.05$, one-way ANOVA with post-hoc Tukey honest significant difference test). *P* values, sample sizes and box plot elements are provided in S1 Table.
(TIF)

**S5 Fig. Response of dark germination mutants to cotylimide-VI (CTL-VI).** Representative images of dark germination mutants germination with or without 2.5 μM CTL-VI.
(TIF)

**S6 Fig. Workflow confirming homozygous *htl-3; pif1* double mutants.** (A) Schematic illustration of WT and *htl-3* allele of *AtKAI2* gene, as well as the primers for *htl-3* genotyping. (B) Schematic illustration of WT and *pif1* T-DNA insertion allele of *AtPIF1* gene, as well as the primers for *pif1* allele genotyping. (C) Confirmation of *htl-3* allele in individual *pif1 htl-3* double mutants. (D) Confirmation of *pif1* allele in individual *pif1; htl-3* double mutants.
(TIF)

**S1 Table. Supporting Table of statistical values.**
(XLSX)

**S2 Table. Supporting Table of Gene Expression Data in Figs 3A, 3B, and 4B.**
(XLSX)

**S3 Table. Supporting Tables of pif1 and KAI20X + KAR$_2$ Gene Expression Data in Fig 3C.**
(XLSX)

**S4 Table. Supporting Table of pif1 and KAI20X + KAR$_2$ Gene Expression Data in Fig 4A.**
(XLSX)

**S5 Table. Supporting Table of germination percentages of Col-0 mutant screen.**
(XLSX)

**S6 Table. Supporting Table primers used in this study.**
(XLSX)

**S7 Table. Supporting Table of genotypes used in this study.**
(XLSX)

## Acknowledgments

We thank Dr. David Nelson for the gift of *smax1-2* and *smax1-2; smxl2-1*. We acknowledge the contribution of Dr. Peter McCourt for discussion and feedback on our manuscript. We acknowledge resource centers like the Arabidopsis Biological Resource Center (ABRC) for providing *Arabidopsis* accessions and mutant lines.

## Author Contributions

**Conceptualization:** Michael Bunsick, Shelley Lumba.

**Data curation:** Jenna E. Hountalas, Gianni Pescetto, George Ly.

**Formal analysis:** Jenna E. Hountalas, Michael Bunsick, Zhenhua Xu, Andrea A. Taylor, George Ly, Shelley Lumba.

**Funding acquisition:** Shelley Lumba.

**Investigation:** Jenna E. Hountalas, Michael Bunsick, Zhenhua Xu, Andrea A. Taylor, Gianni Pescetto, George Ly, Shelley Lumba.

**Methodology:** Jenna E. Hountalas, Michael Bunsick, Zhenhua Xu, Gianni Pescetto, Shelley Lumba.

**Project administration:** Shelley Lumba.

**Resources:** François-Didier Boyer, Christopher S. P. McErlean, Shelley Lumba.

**Software:** George Ly.

**Supervision:** Michael Bunsick, Shelley Lumba.

**Validation:** Michael Bunsick, Zhenhua Xu, George Ly.

**Visualization:** George Ly, Shelley Lumba.

**Writing – original draft:** Jenna E. Hountalas, George Ly, Shelley Lumba.

**Writing – review & editing:** Jenna E. Hountalas, Michael Bunsick, Andrea A. Taylor, George Ly, François-Didier Boyer, Christopher S. P. McErlean, Shelley Lumba.

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
