## [Decision Letter · Decision Letter 0]

30 Mar 2024

Dear Dr Lumba,

Thank you very much for submitting your Research Article entitled 'HTL/KAI2 signalling substitutes for light to control plant germination' to PLOS Genetics.

The manuscript was fully evaluated at the editorial level and by independent peer reviewers. The reviewers appreciated the attention to an important problem, but raised some substantial concerns about the current manuscript. Based on the reviews, we will not be able to accept this version of the manuscript, but we would be willing to review a much-revised version. We cannot, of course, promise publication at that time.

Should you decide to revise the manuscript for further consideration here, your revisions should address the specific points made by each reviewer. We will also require a detailed list of your responses to the review comments and a description of the changes you have made in the manuscript. Please pay particular attention to the germination experiment with far-red light suggested by reviewer #3, and improving explanations of experiments and to referencing/integration of prior research as requested by all three reviewers.

If you decide to revise the manuscript for further consideration at PLOS Genetics, please aim to resubmit within the next 60 days, unless it will take extra time to address the concerns of the reviewers, in which case we would appreciate an expected resubmission date by email to plosgenetics@plos.org.

Please do not hesitate to contact us if you have any concerns or questions.

Yours sincerely,

Caroline Gutjahr

Guest Editor

PLOS Genetics

Claudia Köhler

Section Editor

PLOS Genetics

Reviewer's Responses to Questions

**Comments to the Authors:**

Reviewer #1: This manuscript by Hountalas, Bunsick, Xu et al represents a solid step forward in our understanding of how KAI2/HTL signalling interacts with light in the control of seed germination. The data are resented in a logical manner and the genetic data in particular are very clear, as is the striking similarity in transcriptomes when comparing pif1 and KAR-treated KAI2-OX. Satisfyingly, the data sense from an ecological perspective as well. Liquid light indeed!

I don't have any concerns with the presented data or experiments as described. My main concerns are limited to points that could probably be addressed in the Discussion:

1. Why does 20 µM PAC inhibit the germination of smax1 smxl2 effectively in the dark (Fig 4D), but not smax1 single mutants in the light (Fig 5)? This group reported that smax1 mutants can bypass the germination requirement for GA in the light (Bunsick Nature Plants). But these latest data imply that smax1 cannot do so in the dark. Mechanistically speaking, why? Do SMAX1/SMXL2 vs GA have a relatively different effect on DELLA levels in light vs dark?

2. Perhaps related to the above query, in the Discussion the authors say "Low nanomolar KAR2 concentrations were sufficient to stimulate high germination rates of AtKAI2OX seed in the dark, which was unexpected given that KAR2 addition to AtKAI2OX seed in the light only partially germinates seeds depleted of GA" (lines 357-358). Perhaps I am missing something obvious, but I don't see the link between the two parts of that sentence - unless the surprising fact (or implication) is that GA is required for germination in the dark, but not the light (as stated in the next sentence)? Maybe rearranging the start of this paragraph, and explaining in a little more depth, would help.

3. To this end, I think readers (and myself!) would value a figure that helps to integrate current knowledge of how GA, KAR-KL-SMAX, DELLAs, PIFs and light feed in to the decision to germinate. If desired, such a model could take into account the authors' prediction that "SMAX1/SMXL2 is also necessary for PIF1 function in the dark" (line 337).

4. I was surprised not to find any discussion of Xu et al. (2023) [Cell Reports 10.1016/j.celrep.2023.112740]. This article seems relevant in the context of SMAX1 signalling via DELLAs. Of note, they did not report any germination of (max2) smax1 smxl2 seeds in darkness (I assume different treatment methodologies), although the triple mutant does germinate more readily under low light and the double is relatively more resistant to PAC.

4. Methods Line 444: "Seeds for dark germination assays were left to dry for at least six months before use." Evidently seed storage protocol is an important aspect of this research. Could the authors please explain in more detail how this drying process was done? The seed were harvested first I assume (or were they left to dry on the plants for six months??), then placed in some sort of container (what sort), at low RH (how)?

5. Methods Line 517, the reference gene was apparently AtACTIN8, but in the legend for Fig 1, it says 18S rRNA. Please check and correct. Neither of these is a particularly good reference gene but the changes in gene expression are considerable and unlikely to be affected meaningfully either way. If they were both used, all the better.

Minor concerns

On reading, I noticed a couple of typographical/autocorrect/formatting errors:

Line 57, Striga is a genus, not a genera

Line 321 "SMAX1/SMXL2 inhibits" - I know what you mean, and this feels a bit pedantic, but the use of a slash and singular verb implies that SMAX1 and SMXL2 are the same thing - and they aren't quite. How about "SMAX1 and SMXL2 inhibit..." Likewise lines 104, 197 and probably elsewhere as well.

Line 363, seedlings "squelch" PIF proteins? That's a good one. Sequester?

Line 503, assemblY

Reviewer #2: The manuscript by Hountalas et al is divided into two parts: a first part with elegant seed germination assays (in the dark) and transcriptomic data (see my comments below) leads to the conclusion that the AtKAI2/SMAX1 pathway can overcome the light requirement for Arabidopsis seed germination in the dark; a second part (curiously not mentioned in the abstract) concerns the development of a mutant screen for seed germination in the dark, which allowed the isolation of a novel smax1 mutant (among other uncharacterized mutants).

The manuscript is generally well written, but suffers from a few shortcuts and unexplained points (see my comments). The figures are clear, with a few exceptions (see my comments). My main criticism concerns the use of poorly referenced transcriptomic data, which weakens the message (see below). The discussion is quite interesting

Major points:

-1-Line 135-136: why saying “well characterized lines”? In ref 10 (Bunsick et al, 2020) a single AtKAI2OX line is mentioned (Fig 4); may be cite another reference if both have been described elsewhere? Also, mention which line is shown in Fig1A and D, and used for transcriptomic analysis.

-2-Reference to citation (18) (Oh et al, 2004), occurs repeatedly (lines 94, 96, 216, 228, 234, 236, 241…), but not appropriately to my opinion.

Unless I missed some figures/tables, and the authors did the transcriptomic analysis of WT vs pif1, this citation (18) is provided as reference for the list of PIF1-regulated genes, for Figure 3 and Figure 4 results (Comparison of Differentially Expressed genes versus AtKAI2OX + KAR2). However, this is confusing, because PIF1 regulated genes are not described in Oh et al, 2004. Therefore, it should be clarified where do pif1 transcriptome data come from.

-3-In addition to the precise origin of the pif1 transcriptomic data, it would be useful to indicate the threshold used to distinguish the 61 genes mentioned line 215; this is not obvious to me, even from S3 table.

-4-Line 178-180: for non-specialists and in general for clarity, the use of racGR24 should be explained, and in particular why it is “expected” that the YFP-SMAX1 protein does not accumulate upon the addition of “an artificial SL”. Why not using karrikins? Please indicate the duration of racGR24 treatment.

Also, did the authors test the germination of seeds from the YFP-SMAX1 overexpressing line?

-5-Line 204-206: Is the reference (24) (Huq et al, 2004) appropriate to support the claim?

-6-Figure 5, and line 285 clustering of candidate mutants: The rationale for testing PAC + racGR24 should be clarified. Why not using KAR2?

-7-Line 300, please clarify the sentence, or provide another citation: in ref (32) Cotylimide-VI is described as perturbing SL accumulation, but no mention is made of ABA/GA ratios. Why would “SL constitutive mutants” be found if the screen is made on seed germination (KAI2 pathway)?

To note, the Bleaching/CTL-VI sensitivity of WT is not indicated on Fig 5 (mentioned line 302).

-8-Line 353: which ligand is it about? racGR24 has been used but what is known about SL or KL/KAR sensitivity to light?

-9-Line 368-369: it is not very clear to me what the authors want to say here.

-10-The Abstract last sentence is very attractive, but does not really reflect the work presented.

Minor points:

-Line 250-251, please clarify the sentence (“and consequently SOM”?).

-Line 330: remove “by”

-Line 372: about the fact that light likely stimulates AtKAI2 expression: is it possible to find data that confirm this? in Waters et al Plant Cell 2012, KAI2 transcript levels in seedlings are strongly induced by light.

-Lines 441 and 460: probably refer to another work? Please correct.

-Line 143: please cite a reference for these genes “known to be induced by AtKAI2 activation”.

-Line 156 (legend to figure 1) and methods: How many seeds per replicate? I could not find the information in tables nor in material and methods.

-Line 158 (legend to figure 1): the method section mentions the use of ACTIN 8 (line 517), not 18S rRNA, please correct.

-Line 238: why is SOM qualified as “novel”?

Reviewer #3: In this manuscript by Hountalas et al., it is shown that SMAX1 and SMXL2 repress germination of Arabidopsis thaliana in near darkness. The authors make several interesting new observations including 1) stability of SMAX1 protein in darkness, 2) contribution of SMXL2 to germination, and 3) genetic relationship between htl and pif1. The genetic screen for germination in darkness (and categorization with chemical profiling) is interesting and seems likely to yield interesting new genes. The dark germination method will be a useful technical advance for the strigolactone and karrikin signaling field.

My comments are below:

The manuscript should try to better integrate prior research on KAR signaling, light, and GA during germination. Some of the findings here (as a couple examples, regulation of GA3 oxidases and light-regulated genes in darkness, enhancement of germination by KARs under highly light-limited conditions) by KAR signaling overlap with those made in prior studies but this is not acknowledged or discussed with any meaningful depth.

Some species have positive germination responses to light; others are negatively photoblastic. This is a separate feature from KAR-responsiveness, as some negatively photoblastic species are stimulated by KARs. This should be considered in the discussion and elsewhere (for example, line 43 implies that all "generalist" plants respond positively to light).

The authors should perform a germination experiment with far-red light treatment of Col-0 and smax1 smxl2 to determine whether they have achieved true darkness. Although the light during plating was dim (line 467), there was still light. In addition, phytochrome could have been activated in dry seed during storage or between storage and plating; while this may not have had much effect on wildtype seed, the smax1 smxl2 seed are predisposed to germination and have a lesser light requirement. Without doing this experiment, statements such as "activating the AtKAI2 pathway in the dark completely bypasses the requirement of light for germination in Arabidopsis" (line 79-80) will need to be less bold.

In construction of the double mutant htl-3 pif1-1 (line 460), how was it verified that ShHTL7OX transgene was absent?

The smax1-2; 35S::eYFP-SMAX1 line was analyzed by confocal microscopy in the T2 generation, which is segregating for the transgene. Was this a single-insertion transgene? Was PAC and hygromycin treatment used prior to imaging (line 523)? If so, is it certain that these treatments do not have an effect on SMAX1 stability? Were negative controls (non-transgenic segregants in the same pool) imaged as well? How were they distinguished?

For the mutant screen (line 530), provide details on the EMS treatment (e.g. concentration and duration) and approximate number of M1 plants, which will have implications for the degree of saturation of the screen.

line 68 - clarify "natural SL enantiomers"

line 71 - what qualifies as a "minor role" in germination?

line 93-94 - how does decreasing the level of ABA promote dormancy? this seems backwards

line 117 - is soil devoid of nutrients?

Show evidence in the supplement that benomyl has no effect on germination or seedling viability (line 126-127).

line 147 - this does not logically follow line 146; there may be other ways to stimulate germination in the light than SMAX1/SMXL2 inactivation even if SMAX1/SMXL2 repress germination in the dark

Figure 1B - Is the minimal KAR2 concentration on the x-axis 0.1 nM or is it supposed to be 0? Either way, the data points for this minimal concentration are missing. (Presumably they exist based on the curve fitting)

Various figures - It is not appropriate to use boxplots for small sample sizes (e.g. n=3). There is insufficient data to determine the quartiles for a boxplot. Instead, show the individual data points and the summary statistic (e.g. mean, optionally with SD or SEM bars). For Figure 1D, it is also inappropriate to claim a sample size of 9 (Table S1). From the methods, it sounds like there were three biological samples that were assayed by qPCR with three technical replicates. Use the average of the three technical replicates for each biological sample and show the data for the n=3 biological samples.

line 180 - rac-GR24 is more than just an artificial SL; its component stereoisomers have different activities

line 233 - "nexus for dialling up the combination of hormonal responses" is unclear; please rephrase

PAC application blocked germination of smax1 smxl2 (line 263); discuss why this result is different from Bunsick et al., 2020, Nature Plants "SMAX1-dependent seed germination bypasses GA signalling in Arabidopsis and Striga"

line 323 - it is inaccurate to say that the role of AtKAI2 signaling in germination is "only uncovered when seeds are artificially depleted for GA"; there are many studies which have shown the role of this pathway in germination without depleting GA

line 328-329 - this is confusing; consider rephrasing

line 372 - while it is of course impossible to say at this time how light affect KL production, I suspect there is published data on how light affects AtKAI2 transcript or protein abundance

line 421-425 - I disagree with the validity of these statements and this analysis.

**Have all data underlying the figures and results presented in the manuscript been provided?**

Reviewer #1: Yes

Reviewer #2: **No: **Unless I missed data/tables, see my comment n°1-3

Reviewer #3: Yes

PLOS authors have the option to publish the peer review history of their article (what does this mean?). If published, this will include your full peer review and any attached files.

Reviewer #1: **Yes: **Mark Waters

Reviewer #2: No

Reviewer #3: No

---

## [Decision Letter · Decision Letter 1]

17 Sep 2024

Dear Dr Lumba,

Thank you very much for submitting your Research Article entitled 'HTL/KAI2 signalling substitutes for light to control plant germination' to PLOS Genetics.

The manuscript was fully evaluated at the editorial level and by independent peer reviewers. The reviewers appreciated the findings and your responses to prior reviewer comments.

We therefore ask you to modify the manuscript according to the review recommendations. Your revisions should address the specific points made by each reviewer.

To resubmit, log into your Editorial Manager account and select the option 'Revise Submission' in the 'Submissions Needing Revision' folder.

Yours sincerely,

Caroline Gutjahr

Guest Editor

PLOS Genetics

Claudia Köhler

Section Editor

PLOS Genetics

Reviewer's Responses to Questions

**Comments to the Authors:**

Reviewer #1: In the revised manuscript, the authors have addressed my concerns to my satisfaction. I appreciate the new Figure 6 and the mechanistic explanation for the differential effects of PAC in light vs dark. I think the comments raised by the other reviewers have also been well addressed, and the additional control experiments are welcome. There is one, however, that I think remains to be managed.

Reviewer 3 queried the logic of lines 421-425 (now lines 485-489). The first time I read this section, I also paused for thought because something seemed amiss - the analysis seemed a little too simplistic. At the time I didn't bother thinking it through properly. But seeing as reviewer #3 raised it but did not specify exactly what the problem was, causing the authors to dismiss the concern, I thought I would revisit it.

I suppose the problem is concluding "the probability of finding two nonsense alleles and zero missense alleles" to be 1/125. This calculation (0.089^2 expressed as an approximate fraction) identifies the likelihood that, if two EMS-mutable sites in SMAX1 are mutated at random, both would result in a stop codon. It doesn't say anything about the probability of recovering these alleles (and not others) in a mutant screen, which is what the authors claim. The two known nonsense alleles were identified in two different screens. To say anything about the present screen being "impervious" to identifying missense mutations, you would surely need to recover nonsense alleles multiple times over, and calculate the probability of that outcome, given the likelihood that the other 91% of sites would also be hit. With just a single "hit" and no saturation, you can't assess probabilities like that. And what proportion of the other 91% of sites would give a neutral missense allele vs a detrimental one? This is impractical to calculate. It is normal for mutant screens to identify strong alleles - missense alleles are likely to be weaker than nonsense – so I'm not really sure what point the authors are trying to make here. I think this section could be removed without impacting the work.

Reviewer #3: Point #1 - I was asking the authors to more thoroughly cite and discuss in the manuscript the findings in the preexisting literature that are relevant to this topic (e.g. KAR signaling, light, and GA during germination). The authors' response to this point seems to be intended for a different comment.

Point #2 - Please indicate in the manuscript that not all species are positively photoblastic and provide a reference to support the claim that the majority of plants use light as a positive germination signal (while making this claim, the authors may wish to keep in mind that light signals are relevant to seeds with physiological dormancy and that several other forms of seed dormancy exist).

Point #3 - I appreciate the experiment with far-red light being performed. It remains possible that the 5 minute far-red light treatment was of insufficient intensity and duration to turn off phytochrome (this could be tested by applying the far-red treatment, or not, after a red light treatment) but if the authors are comfortable with the experimental setup it is fine with me.

Point #21 - In the previous manuscript's lines 421-425, the authors wrote "Given that only 8.9% of the EMS mutable sites in SMAX1 produce a nonsense codon, the probability of finding two nonsense alleles (smax1-1 and smax1-4) and zero missense alleles is approximately 1/125. This suggests that our screen is relatively impervious to identifying SMAX1 missense mutations possibly due to the stringency of dark as a germination condition." As I stated before, I do not agree that this is a valid analysis or conclusion.

Leaving aside the implicit assumption that EMS only causes transition point mutations, which is not true, this calculation assumes sampling mutants from an M2 pool in which all possible transition mutations in SMAX1 are represented and that each mutant allele is equally abundant in the pool being screened. Neither of these conditions are likely to be true, not least because EMS mutagenesis is not truly random (genome-wide bias based on chromatin and nearby sequence context have been observed) and because other second-site mutations in the genome of any given mutant frequently affect fertility/viability. It also assumes that the 91.1% of other EMS-mutable sites in SMAX1 coding sequence can produce missense mutations. In reality, only a subset would do so and many EMS mutations will be silent. So the correct comparison, again assuming equal representation, would adjust the proportion of possible nonsense mutations among the population of possible nonsense and missense mutations combined. Finally, these two alleles represent a small number of very infrequent occurrences among a large M2 population. In such cases, a Poisson distribution is more appropriate for inferring probability (not that I recommend doing so here).

Reviewer #4: This manuscript details the important role of SMAX1 in Arabidopsis germination in the dark and provides evidence that both SMAX1 and SMXL2 are necessary for PIF1 function under these conditions. A working model is presented to describe the contributions of SMAX1 and SMXL2 to germination in dark versus light conditions, effectively integrating prior research on KAR signaling, light, and GA during germination. These results are significant and impactful in explaining how specialized plants, such as ephemeral and parasitic weeds, have evolved their germination behavior in response to specific environments.

Major comments：

1.Fig 5, the photographs showing the bleached seedling phenotype of various mutants grown on 2.5 µM CT-VI are not clear enough, possibly due to insufficient image clarity or small size. So I don't think it is obvious that "mutants constitutive for KL signaling, like smax1-2, may show increased sensitivity to CTL-VI application compared to ABA auxotrophs with respect to cotyledon bleaching." Please present the results of CTL-VI treatment on the mutants in a separate figure.

2.Why was the expression of genes known to be induced by AtKAI2 activation (DLK2, KUF1, BBX20) lower in dark-germinated AtKAI2OX-A seeds treated with DMSO than in wild-type seeds (Fig 1D)?

3.Could the author please explain how to treat smax1-2; smxl2-1 seed with Abscisic acid (ABA) and paclobutrazol (PAC)（S4 Fig）？

4.Please explain why smax1-4 germinated well in the dark while smax1-2 did not germinate well in the dark (Fig 5).

5.Line 423-424 (Fig 6) in the Revised Article with Changes Highlighted version: Which EAR repressors are recruited by SMAX1 and SMXL2 proteins? Please provide suitable references.

6.Why are smax1-2; 35S::eYFP-SMAX1 seeds sprinkled onto ½ MS (Murashige-Skoog) agar (0.8%), while the other seeds in the Dark Germination Assays are sprinkled directly onto Petri plates in the dark containing water agar (0.8%) and the fungicide benomyl (10 µg/ml)?

Minor comments：

1.Line 188 of the Revised Article with Changes Highlighted version: The reference to "S2 Fig" is incorrect. "S2 Fig" does not mention PAC treatment.

2.Line 277：S4 Fig，not S3 Fig.

3.Line 327: "Mutants perturbed in KL signaling can be identified using a compound called cotylimide-VI (CTL-VI) [36].” Should this reference be to citation 37 instead of 36? Please confirm.

4.Line 502：“Col-0” should not be written in italics.

5.Line 503: “smax1-2/smxl2-1 [38] pif1-1 [21]”, a comma is missing here.

6.Line 504:“abi4-5 [53], cop1-4 [53], aba2-2 [54], abi1-1 [51], det1-1 [55]”All genotypes in this sentence should be italicized.

7.Line 506: Please clarify the sentence, “All seeds were grown in a continuous 24-hour light chamber at 22°C.” What kind of photoperiod does Arabidopsis thaliana require for growth?

8.Line 509:“240 C”The unit of Celsius is misspelled.

9.Line 507-509：“Seeds for dark germination assays were left to dry for at least six months in a glass tube and closed cardboard storage box at room temperature (240 C) in the laboratory before use.”This sentence should be moved to the method part of “Dark Germination Assays”.

10.Line 535-537:“Approximately 25 to 35 of unsterilized dry seeds per replicate were sprinkled directly onto 10 mm petri plates in the dark containing water agar (0.8%) and the fungicide benomyl (10 µg/ml) (Toronto Research Chemicals, B161380). Unsterilized dry seeds were sprinkled directly onto 10 mm petri plates in the dark containing water agar (0.8%) and the fungicide benomyl (10 µg/ml) (Toronto Research Chemicals, B161380).” This passage contains a repeated sentence.

**Have all data underlying the figures and results presented in the manuscript been provided?**

Reviewer #1: Yes

Reviewer #3: Yes

Reviewer #4: None

PLOS authors have the option to publish the peer review history of their article (what does this mean?). If published, this will include your full peer review and any attached files.

Reviewer #1: No

Reviewer #3: No

Reviewer #4: No

---

## [Editor Report · Decision Letter 2]

3 Oct 2024

Dear Dr Lumba,

We are pleased to inform you that your manuscript entitled "HTL/KAI2 signalling substitutes for light to control plant germination" has been editorially accepted for publication in PLOS Genetics. Congratulations!

Yours sincerely,

Caroline Gutjahr

Guest Editor

PLOS Genetics

Claudia Köhler

Section Editor

PLOS Genetics

Comments from the reviewers (if applicable):

**Data Deposition**

http://datadryad.org/submit?journalID=pgenetics&manu=PGENETICS-D-24-00244R2

**Press Queries**

---

## [Editor Report · Acceptance letter]

16 Oct 2024

PGENETICS-D-24-00244R2 

HTL/KAI2 signalling substitutes for light to control plant germination 

Dear Dr Lumba, 

We are pleased to inform you that your manuscript entitled "HTL/KAI2 signalling substitutes for light to control plant germination" has been formally accepted for publication in PLOS Genetics! Your manuscript is now with our production department and you will be notified of the publication date in due course.

With kind regards,

Anita Estes

PLOS Genetics

On behalf of:
